# Açaì (*Euterpe oleracea*) Extract Protects Human Erythrocytes from Age-Related Oxidative Stress

**DOI:** 10.3390/cells11152391

**Published:** 2022-08-03

**Authors:** Alessia Remigante, Sara Spinelli, Elisabetta Straface, Lucrezia Gambardella, Daniele Caruso, Giuseppe Falliti, Silvia Dossena, Angela Marino, Rossana Morabito

**Affiliations:** 1Department of Chemical, Biological, Pharmaceutical and Environmental Sciences, University of Messina, 98166 Messina, Italy; aremigante@unime.it (A.R.); sspinelli@unime.it (S.S.); marinoa@unime.it (A.M.); 2Institute of Pharmacology and Toxicology, Paracelsus Medical University, 5020 Salzburg, Austria; silvia.dossena@pmu.ac.at; 3Biomarkers Unit, Center for Gender-Specific Medicine, Istituto Superiore di Sanità, 00161 Rome, Italy; elisabetta.straface@iss.it (E.S.); lucrezia.gambardella@iss.it (L.G.); 4Complex Operational Unit of Clinical Pathology of Papardo Hospital, 98166 Messina, Italy; daniele.caruso1985@libero.it (D.C.); peppefal@tin.it (G.F.)

**Keywords:** Açaí berry, d-Galactose, aging, oxidative stress, glycation, plasma membrane, band 3 protein function, erythrocytes

## Abstract

Aging is a process characterised by a general decline in physiological functions. The high bioavailability of reactive oxygen species (ROS) plays an important role in the aging rate. Due to the close relationship between aging and oxidative stress (OS), functional foods rich in flavonoids are excellent candidates to counteract age-related changes. This study aimed to verify the protective role of Açaì extract in a d-Galactose (d-Gal)-induced model of aging in human erythrocytes. Markers of OS, including ROS production, thiobarbituric acid reactive substances (TBARS) levels, oxidation of protein sulfhydryl groups, as well as the anion exchange capability through Band 3 protein (B3p) and glycated haemoglobin (A1c) have been analysed in erythrocytes treated with d-Gal for 24 h, with or without pre-incubation for 1 h with 0.5–10 µg/mL Açaì extract. Our results show that the extract avoided the formation of acanthocytes and leptocytes observed after exposure to 50 and 100 mM d-Gal, respectively, prevented d-Gal-induced OS damage, and restored alterations in the distribution of B3p and CD47 proteins. Interestingly, d-Gal exposure was associated with an acceleration of the rate constant of SO_4_^2−^ uptake through B3p, as well as A1c formation. Both alterations have been attenuated by pre-treatment with the Açaì extract. These findings contribute to clarify the aging mechanisms in human erythrocytes and propose functional foods rich in flavonoids as natural antioxidants for the treatment and prevention of OS-related disease conditions.

## 1. Introduction

Aging is a dynamic chronological process characterized by the gradual accumulation of damage to cells, progressive functional decline, and increased susceptibility to disease [1]. A causal hypothesis that gained considerable interest in recent years postulates that pathophysiological changes during aging are due to progressive oxidative damage to cellular macromolecules [2,3,4,5,6]. In physiological conditions, the production of reactive oxygen and nitrogen species generated during cellular metabolism in biological systems is balanced by the ability of the latter to defend through their sophisticated antioxidant machinery. Nevertheless, when oxidants are produced in excess, or when the antioxidant defenses that regulate them are ineffective, this balance can be perturbed, thus resulting in oxidative stress (OS) [4,7,8,9,10]. In these conditions, biomolecules can be altered through oxidation to an extent that exceeds repair capacity. Specifically, OS induces lipid peroxidation and glycoxidation reactions, which lead to the formation of highly reactive species that attack free amino groups in proteins, causing their covalent modifications, thus resulting in the generation of advanced glycation and lipo-oxidation end products (AGEs and ALEs) [11,12]. AGEs impair the protein structure due to covalent cross-linking, resulting in protein oligomerization and aggregation. These modifications lead to alterations in the cellular structure and/or function, and ultimately cell death through apoptosis or necrosis. Abnormal levels of reactive species may be the common denominator underlying aging in several acute and chronic pathologies, including systemic sclerosis, cardiovascular diseases, or chronic obstructive pulmonary disease, although the specific mechanisms contributing to OS-induced damage are poorly investigated [13].

Although many cellular models have been used to study the biochemical alterations during aging, erythrocytes get superiority amongst them [3]. In fact, red blood cells are unique, highly specialized, and the most abundant cells in different organisms [14]. Although their primary function is the transportation of the respiratory gases O_2_ and CO_2_ between the lungs and tissues, these circulatory cells are equipped with effective anti-oxidative systems that make them mobile free radical scavengers providing antioxidant protection not only to themselves, but also to other tissues and organs in the body [15]. In fact, erythrocytes can contribute to detoxify reactive species, thus rescuing or partially protecting cells in regions of increased OS. However, erythrocytes face particular metabolic challenges regarding the generation and mitigation of oxidative damage. First, erythrocytes have a unique source of OS because of the high load of iron associated with haemoglobin, which drives radical-generating Fenton reactions [16]. Second, unlike most other tissues, the repair of OS in erythrocytes cannot involve de novo protein synthesis owing to the lack of nuclei and ribosomes. As such, elucidating the pathways by which red blood cells counteract OS can provide unique clues on cellular responses to oxidant injury [17,18]. Alterations in morphological properties of erythrocytes, such as adhesivity, aggregability, and deformability, have been detected in aging, as well as a number of human pathologic conditions displaying systemic OS as a hallmark [19].

During aging, the erythrocyte volume decreases with time, with an increase in density and a decrease in the haemoglobin content, especially during the first and the second part of the lifespan, respectively. These changes are associated with a loss of cholesterol and phospholipids and a decrease in the mean surface area of 20%, thus indicating a loss of the membrane constituents during aging. Membrane loss occurs through the formation of vesicles containing haemoglobin, haem–Fe, and iron, which act as oxidants able to modulate blood functions [20]. Consequently, aging erythrocytes lose the membrane domain, their normal deformability, and consequently, their classic biconcave disc shape. Moreover, phosphatidylserine, which is normally exposed to the intracellular side of the plasma membrane, becomes externalised [21]. The removal of erythrocytes from circulation is known to be accelerated by age-related changes in cell membrane composition, which, in turn, alters the rheological and immunological properties of these cells [22].

Functional foods and natural products with antioxidant properties can potentially improve the life functions of erythrocytes and, consequently, the homeostasis of the whole body. Euterpe oleracea, an Amazonian Brazilian fruit popularly known as Açaì [23], presents several bioactive molecules with different properties, including antioxidant, anti-inflammatory, and analgesic activities, and also modulates calcium homeostasis. The biological effects of Açaì are related to its chemical matrix, which includes numerous phytochemicals components, such as flavonoids [24]. These molecules can directly neutralize reactive oxygen species (ROS) and/or inactivate molecules with a pro-oxidant capacity [25]. Since aging of the circulating erythrocytes is tightly related to OS, we hypothesized that an Açaì extract might have beneficial effects against aging-related processes in these cells. Among the experimental models of aging, long-term d-Galactose (d-Gal) exposure is the most similar to natural aging [26,27]. Here, we explore the possible protective effect of an Açaì extract on age-related events such as oxidative damage and glycation in a d-Gal-induced model of aging in human erythrocytes.

## 2. Materials and Methods

### 2.1. Solutions and Chemicals

All chemicals were purchased from Sigma (Milan, Italy). 4,4′-diisothiocyanatostilbene-2,2′-disulfonate (DIDS) stock solution (10 mM) was prepared in dimethyl sulfoxide (DMSO). d-Galactose stock solution (1 M) was prepared in distilled water. Freeze-dried Açaì extract was dissolved in distilled water. This substance (Cas Number: 879496-95, 906351-38-0) was originally purchased from Farmalabor Srl (Canosa di Puglia, Barletta, Italy), and then kindly provided to us by Professor M. Cordaro from University of Messina. N-ethylmaleimide (NEM) (310 mM) stock solution was prepared in ethanol. H_2_O_2_ was diluted in distilled water from a 30% *w*/*w* stock solution. Ethanol never exceeded 0.001% *v*/*v* in the experimental solutions and was previously tested on erythrocytes to exclude haemolysis.

### 2.2. Erythrocyte Preparation

This study was prospectively reviewed and approved by a duly constitute Ethics Committee (prot.52-22, 20-04-2022). Upon informed consent, whole human blood from healthy volunteers was collected in test tubes containing ethylenediaminetetraacetic acid (EDTA). Plasma concentration of glycated haemoglobin (A1c) was less than 5%. Erythrocytes were washed in isotonic solution (composition in mM: NaCl 150, 4-(2-hydroxyethyl)-1-piperazineethanesulfonic acid (HEPES) 5, Glucose 5, pH 7.4, osmotic pressure 300 mOsm/kgH_2_O) and centrifuged thrice (Neya 16R, 1200× *g*, 5 min) to remove plasma and buffy coat. Erythrocytes were then suspended at specific haematocrits in isotonic solution and addressed to downstream analysis.

### 2.3. Haemolysis Measurement

To verify the % haemolysis, erythrocytes (35% haematocrit) were incubated with or without Açaì extract in isotonic solution, suspended at 0.5% haematocrit in isotonic solution, centrifuged (Neya 16R, 1200× *g*, 5 min), and resuspended at 0.05% haematocrit in a 0.9% *v*/*v* NaCl solution [28,29]. Haemoglobin absorbance was measured at 405 nm wavelength and subtracted for the absorbance of blank (0.9% *v*/*v* NaCl solution).

### 2.4. Thiobarbituric-Acid-Reactive Substances (TBARS) Level Measurement

TBARS levels were measured as described by Mendanha and collaborators [30], with minor modifications. TBARS are derived from the reaction between thiobarbituric acid (TBA) and malondialdehyde (MDA), which is the end-product of lipid peroxidation [30]. Erythrocytes were suspended at 20% haematocrit and pre-incubated in the presence or absence of different concentrations of Açaì extract for 1 h at 37 °C. Successively, samples were incubated with 50 or 100 mM d-Gal for 24 h at 25 °C. Then, samples were centrifuged (Neya 16R, 1200× *g*, 5 min) and suspended in isotonic solution. Erythrocytes (1.5 mL) were treated with 10% (*w*/*v*) trichloroacetic acid (TCA) and centrifuged (Neya 16R, 3000× *g*, 10 min). TBA (1% in hot distilled water, 1 mL) was added to the supernatant and the mixture was incubated at 95 °C for 30 min. At last, TBARS levels were obtained by subtracting 20% of the absorbance at 453 nm from the absorbance at 532 nm (Onda Spectrophotometer, UV-21). Results are indicated as µM TBARS levels (1.56 × 10^5^ M^−1^ cm^−1^ molar extinction coefficient).

### 2.5. Total Sulfhydryl Group Content

Measurement of total -SH groups was carried out according to the method of Aksenov and Markesbery [31], with minor modifications. In short, erythrocytes (35% haematocrit), left untreated or exposed to d-Gal-containing solutions with or without pre-incubation with Açaì extract, were centrifuged (Neya 16R, 1200× *g*, 5 min) and 100 µL haemolysed in 1 mL of distilled water. A 50 μL aliquot was added to 1 mL of phosphate-buffered saline (PBS, pH 7.4) containing EDTA (1 mM). 5,5′-Dithiobis (2-nitrobenzoic acid) (DTNB, 10 mM, 30 μL) was added to initiate the reaction and the samples were incubated for 30 min at 25 °C, protected from light. Control samples, without cell lysate or DTNB, were processed concurrently. After incubation, sample absorbance was measured at 412 nm (Onda spectrophotometer, UV-21) and 3-thio-2-nitro-benzoic acid (TNB) levels were detected after subtraction of blank absorbance (samples containing only DTNB). To achieve full oxidation of -SH groups, an aliquot of erythrocytes (positive control) was incubated with 2 mM NEM for 1 h at 25 °C [32,33]. Data were normalised to protein content and results reported as μM TNB/mg protein.

### 2.6. Analysis of Cell Shape by Scanning Electron Microscopy (SEM)

Samples, left untreated or exposed to d-Gal-containing solutions with or without pre-incubation with Açaì extract, were collected, plated on poly-l-lysine-coated slides and fixed with 2.5% glutaraldehyde in 0.1 M cacodylate buffer (pH 7.4) at room temperature for 20 min. Then, samples were post-fixed with 1% OsO_4_ in 0.1 M sodium cacodylate buffer and dehydrated through a graded series of ethanol solutions (from 30% to 100%). Then, absolute ethanol was gradually substituted by a 1:1 solution of hexamethyldisilazane (HMDS)/absolute ethanol and successively by pure HMDS. Successively, HMDS was completely removed and samples were dried in a desiccator. Dried samples were mounted on stubs, coated with gold (10 nm), and analyzed by a Cambridge 360 scanning electron microscope (Leica Microsystem, Wetzlar, Germany) [34]. Altered erythrocyte shape was evaluated by counting ≥500 cells (50 erythrocytes for each different SEM field at a magnification of 3000×) from samples in triplicate.

### 2.7. Detection of Reactive Oxygen Species (ROS)

To evaluate intracellular reactive oxygen intermediates, erythrocytes, left untreated or exposed to d-Gal-containing solutions with or without pre-incubation with Açaì extract, were incubated in Hanks’ balanced salt solution, pH 7.4, containing dihydrorhodamine 123 (DHR 123; Molecular Probes, Milan, Italy), and then analyzed with a FACScan flow cytometer (Becton-Dickinson, Mountain View, CA, USA). At least 20,000 events were acquired. The median values of fluorescence intensity histograms were used to provide a semi-quantitative analysis of ROS production [35].

### 2.8. Detection of Apoptotic Erythrocytes

Erythrocytes, left untreated or exposed to d-Gal-containing solutions with or without pre-incubation with Açaì extract, were processed to detect apoptosis by using the FITC-conjugated Annexin V apoptosis detection kit (Biovision, CA, USA.) and Trypan blue staining (0.05% Trypan blue for 10 min at room temperature) [36]. Then, erythrocytes were analyzed with a FACScan flow cytometer (Becton-Dickinson, Mountain View, CA, USA) equipped with a 488 nm argon laser.

### 2.9. Analytical Cytology

Erythrocytes, left untreated or exposed to d-Gal-containing solutions with or without pre-incubation with Açaì extract, were fixed with 3.7% formaldehyde in PBS (pH 7.4) for 10 min at room temperature and then washed in the same buffer. Cells were then permeabilised with 0.5% Triton X-100 in PBS for 5 min at room temperature. After washing with PBS, samples were incubated with monoclonal anti-Band 3 protein (Sigma, Milan, Italy) or monoclonal anti-CD47 (Santa Cruz Biotechnology, Dallas, TX, United States) antibodies for 30 min at 37 °C, washed, and then incubated with a fluorescein isothiocyanate (FITC)-labeled anti-mouse antibody (Sigma, Milan, Italy) for 30 min at 37 °C [20]. Cells incubated with the secondary antibody given alone were used as the negative control. Samples were analyzed by an Olympus BX51 Microphot fluorescence microscope or by a FACScan flow cytometer (Becton Dickinson, Mountain View, CA, USA) equipped with a 488 nm argon laser. At least 20,000 events have been acquired. The median values of fluorescence intensity histograms were used to provide a semiquantitative analysis. Fluorescence intensity values were normalised for those of untreated erythrocytes and expressed in %.

### 2.10. Measurement of Glycated Haemoglobin (%A1c) Levels

The glycated haemoglobin content (%A1c) was determined with the A1c liquidirect reagent as previously described by Sompong and collaborators [37], with minor modifications. Briefly, after incubation with d-Gal with or without pre-incubation with frozen Açaì extract, erythrocytes were lysed in hypotonic buffer and then incubated with latex reagent at 37 °C for 5 min. The absorbance of samples was measured at 610 nm (BioPhotometer Plus, Eppendorf, Manchester, United Kingdom) and the A1c content was calculated from a standard curve constructed by using known A1c concentrations and expressed in %.

### 2.11. SO_4_^2−^ Uptake Measurement

#### 2.11.1. Control Condition

SO_4_^2−^ uptake measurement was used to evaluate the anion exchange through B3p, as described elsewhere [38,39,40,41]. Briefly, after washing, erythrocytes were suspended to 3% haematocrit in 35 mL SO_4_^2−^ medium (composition in mM: Na_2_SO_4_ 118, HEPES 10, glucose 5, pH 7.4, osmotic pressure 300 mOsm/kgH_2_O) and incubated at 25 °C. After 5, 10, 15, 30, 45, 60, 90, and 120 min, DIDS (10 μM), which is an inhibitor of B3p activity [42], was added to 5 mL sample aliquots, which were kept on ice. Subsequently, samples were washed three times in cold isotonic solution and centrifuged (Neya 16R, 4 °C, 1200× *g*, 5 min) to eliminate SO_4_^2−^ from the external medium. Distilled water (1 mL) was added to induce osmotic lysis of erythrocytes and perchloric acid (4% *v*/*v*) was used to precipitate proteins. After centrifugation (Neya 16R, 4 °C, 2500× *g*, 10 min), the supernatant containing SO_4_^2−^ trapped by erythrocytes was directed to the turbidimetric analysis. Supernatant (500 μL from each sample) was sequentially mixed to 500 μL glycerol diluted (1:1) in distilled water, 1 mL 4 M NaCl, and 500 μL 1.24 M BaCl_2_•2H_2_O. Finally, the absorbance of each sample was measured at 425 nm (Spectrophotometer, UV-21, Onda Spectrophotometer, Carpi, Modena, Italy). By means of a calibrated standard curve previously obtained by precipitating known SO_4_^2−^ concentrations, the absorbance was converted to [SO_4_^2−^] L cells × 10^−2^. The rate constant of SO_4_^2−^ uptake (min^−^^1^) was derived from the following equation: C_t_ = C_∞_ (1 − e^−rt^) + C_0_, where C_t_, C_∞_, and C_0_ indicate the intracellular SO_4_^2−^ concentrations measured at time t, ∞, and 0, respectively, e represents the Neper number (2.7182818), r indicates the rate constant accounting for the process velocity, and t is the specific time at which the SO_4_^2−^ concentration was measured. The rate constant is the inverse of the time needed to reach ~63% of total SO_4_^2−^ intracellular concentration [38] and [SO_4_^2−^] L cells × 10^−2^ reported in figures represents SO_4_^2−^ micromolar concentration internalized by 5 mL erythrocytes suspended at 3% haematocrit.

#### 2.11.2. Experimental Conditions

After 1 h pre-incubation with or without freeze-dried Açaì extract at 37 °C, erythrocytes (3% haematocrit) were exposed to d-Gal (50 or 100 mM) for 24 h at 25 °C. Successively, samples were centrifuged (Neya 16R, 4 °C, 1200× *g*, 5 min) to replace the supernatant with SO_4_^2^^−^ medium. The rate constant of SO_4_^2−^ uptake was then determined as described for the control condition.

### 2.12. Experimental Data and Statistics

All data are expressed as arithmetic means ± standard error of the mean. For statistical analysis and graphics, GraphPad Prism (version 8.0, GraphPad Software, San Diego, CA, USA) and Excel (Version 2019, Microsoft, Redmond, WA, USA) software were used. Data normality was verified with the D’Agostino and Pearson Omnibus normality test. Significant differences between mean values were determined by one-way analysis of variance (ANOVA), followed by Bonferroni’s multiple comparison post-test or ANOVA with Dunnet’s post-test, as appropriate. Statistically significant differences were assumed at *p* < 0.05; (*n*) corresponds to the number of independent measurements.

## 3. Results

### 3.1. Effect of Increasing Concentrations of Freeze-Dried Açaì Extract on Haemolysis, Thiobarbituric-Acid-Reactive Substances (TBARS) Levels, and Sulfhydryl Groups Total Content

To select the concentrations of the freeze-dried Açaì extract to apply for downstream analysis, we measured the haemolysis percentage, thiobarbituric acid reactive substances (TBARS) levels, and the total content of sulfhydryl groups following the incubation of erythrocytes with increasing concentrations (0.005–1000 µg/mL) of freeze-dried Açaì extract for 1 h at 37 °C. As shown in Figure 1, 1000 µg/mL of freeze-dried Açaì extract significantly increased the haemolysis percentage compared to cells left untreated (control). As expected, in erythrocytes exposed to 20 mM H_2_O_2_, TBARS levels were significantly higher with respect to those of control erythrocytes (Figure 2A). The Açaì extract in the concentration range between 50 and 1000 µg/mL induced a significant increase of TBARS levels compared to control cells (Figure 2A). Conversely, treatment with 0.005–10 µg/mL did not significantly affect TBARS levels compared to the control (Figure 2A). Figure 2B shows the total content of sulfhydryl groups (µM TNB/µg protein) of erythrocytes left untreated or treated with either the oxidising compound NEM (2 mM for 1 h, as the positive control) or different concentrations of freeze-dried Açaì extract for 1 h at 37 °C. As expected, exposure to NEM led to a significant reduction in sulfhydryl groups content. The Açaì extract in the range from 50 to 1000 µg/mL also significantly reduced sulfhydryl group abundance with respect to control, while lower concentrations were ineffective. Based on these results, only concentrations of freeze-dried Açaì extract <50 µg/mL, which do not induce haemolysis or OS, have been selected for the experiments shown in the following.

### 3.2. Freeze-Dried Açaì Extract Prevents Cell Shape Changes in d-Gal-Treated Erythrocytes

As depicted in Figure 3, treatment with d-Gal (50 or 100 mM) for 24 h induced morphological alterations of erythrocytes. In fact, by scanning electron microscopy analysis we detected 15.7% of acanthocytes (erythrocytes with surface blebs) in samples treated with 50 mM d-Gal and 28.2% of leptocytes (erythrocytes with a flattened shape) in samples treated with 100 mM d-Gal (Table 1). Pre-treatment with 0.5 or 10 µg/mL freeze-dried Açaì extract reduced the percentage of morphologically altered cells. Specifically, in samples treated with 50 mM d-Gal, the percentage of acanthocytes was reduced to 6.9% and 9.8% in samples pre-treated with 0.5 and 10 µg/mL Açaì, respectively. Instead, in samples treated with 100 mM d-Gal, the percentage of leptocytes was reduced to 22% and 12% after pre-treatment with 0.5 and 10 µg/mL Açaì, respectively.

### 3.3. Evaluation of Intracellular ROS Levels

The evaluation of ROS species was carried out by flow cytometry in erythrocytes left untreated or, alternatively, exposed to d-Gal with or without pre-exposure to 0.5 or 10 µg/mL freeze-dried Açaì extract for 1 h. Figure 4 shows the intracellular ROS levels at different time points (0, 3, 5, and 24 h after exposure to d-Gal). Samples exposed to 50 or 100 mM d-Gal showed a significant increase of ROS levels compared to the control samples. After 3 h, levels of ROS increased by 50% in d-Gal treated samples and remained unchanged in time. In Figure 4, the effect of freeze-dried Açaì extract is also reported. In samples pre-exposed to 0.5 or 10 µg/mL Açaì extract, 50 or 100 mM d-Gal failed to significantly increase ROS levels, which remained unchanged compared to control values (Figure 4A,B).

### 3.4. Measurement of Thiobarbituric-Acid-Reactive Substances (TBARS) Levels

Thiobarbituric-acid-reactive substances (TBARS) measurements in erythrocytes are reported in Figure 5. As expected, TBARS levels of erythrocytes treated with 20 mM H_2_O_2_ for 1 h were significantly higher with respect to those of erythrocytes left untreated (control). Similarly, after 24 h of incubation with 50 and 100 mM d-Gal, TBARS levels were significantly increased with respect to those of control erythrocytes. Importantly, in erythrocytes pre-treated with increasing concentrations of freeze-dried Açaì extract, and then exposed to 50 or 100 mM d-Gal, TBARS levels were significantly reduced compared to those measured in 50 or 100 mM d-Gal-treated erythrocytes. Of note, freeze-dried Açaì extracts alone did not significantly affect TBARS levels (Figure 2A).

### 3.5. Measurement of Total Sulfhydryl Group Content

Figure 6 shows the total content of sulfhydryl groups (µM TNB/µg protein) in erythrocytes left untreated or treated with either the oxidising compound NEM (2 mM for 1 h, as the positive control), or 50 and 100 mM d-Gal for 24 h with or without pre-treatment with freeze-dried Açaì extract. As expected, exposure to NEM led to a significant reduction in the sulfhydryl groups’ content. Sulfhydryl groups in 50 and 100 mM d-Gal-treated erythrocytes were also significantly reduced with respect to the control. Importantly, pre-treatment with freeze-dried Açaì extract (0.5 or 10 µg/mL) significantly restored the total content of the sulfhydryl groups in 50 or 100 mM d-Gal-treated erythrocytes (Figure 6). Freeze-dried Açaì extracts alone did not significantly affect the total sulfhydryl groups’ content (Figure 2B).

### 3.6. Determination of Aging Markers

Phosphatidylserine (PS) externalisation, CD47 protein, and B3p have been selected as aging markers. Regarding the percentage of erythrocytes with PS externalisation (apoptosis) no significant difference was detected after treatment with 50 and 100 mM d-Gal (Figure 7).

Regarding CD47, flow cytometry analysis has shown a significantly decreased expression in samples treated with 50 or 100 mM d-Gal for 24 h compared to untreated (control) samples (Figure 8A,B). Pre-treatment (1 h) with 0.5 µg/mL freeze-dried Açaì extract did not restore CD47 expression in 50 mM d-Gal-treated erythrocytes, while a total restoration of CD47 expression was evident in 100 d-Gal-treated erythrocytes. Conversely, the expression of this protein was significantly restored by pre-treatment with 10 µg/mL freeze-dried Açaì in both 50 and 100 mM d-Gal-treated erythrocytes. Freeze-dried Açaì extract alone did not significantly affect CD47 expression (data not shown). Data obtained by flow cytometry were confirmed by immunofluorescence analyses, which showed a dramatic rearrangement and redistribution of this protein (Figure 8C,D).

B3p protein expression was found significantly decreased in human erythrocytes treated with 50 or 100 mM d-Gal for 24 h with respect to those left untreated (control) (Figure 9). Freeze-dried Açaì extract (0.5 or 10 µg/mL) pre-treatment did not restore B3p expression in erythrocytes treated with 50 mM d-Gal (Figure 9A). Conversely, B3p expression was significantly restored in erythrocytes pre-treated with 10 µg/mL freeze-dried Açaì extract (Figure 9). Freeze-dried Açaì extracts alone did not significantly affect B3p expression (data not shown). In addition, a redistribution of B3p was detected by immunofluorescence (Figure 9E). In particular, B3p was mainly localised in blebs (arrows) of acanthocytes after treatment with 50 mM d-Gal, or alternatively, clustered (arrows) in leptocytes after treatment with 100 mM d-Gal, with respect to untreated erythrocytes. In the latter, these changes were attenuated by freeze-dried Açaì extract (10 µg/mL) pre-treatment.

### 3.7. Measurement of Glycated Haemoglobin (%A1c) Levels

Figure 10 shows the glycated haemoglobin levels (%A1c) measured in erythrocytes left untreated or treated with 50 or 100 mM d-Gal for 24 h with or without pre-treatment with 0.5 or 10 µg/mL freeze-dried Açaì extract for 1 h at 37 °C. The %A1c levels measured following exposure to 50 or 100 mM d-Gal were significantly increased with respect to those of erythrocytes left untreated (control). Pre-incubation with freeze-dried Açaì extract (0.5 or 10 µg/mL) for 1 h significantly reduced the %A1c levels in both 50 or 100 mM d-Gal-treated erythrocytes towards values that did not differ from control values. Freeze-dried Açaì extracts alone did not significantly affect the %A1c content (data not shown).

### 3.8. SO_4_^2−^ Uptake Measurement

Figure 11 describes the SO_4_^2−^ uptake as a function of time in erythrocytes left untreated (control) and in erythrocytes treated with 50 or 100 mM d-Gal for 24 h with or without pre-incubation with 0.5 or 10 µg/mL freeze-dried Açaì extract for 1 h at 37 °C. In control conditions, SO_4_^2−^ uptake progressively increased and reached equilibrium within 45 min (rate constant of SO_4_^2−^ uptake = 0.059 ± 0.001 min^−1^). Erythrocytes treated with 0.5 or 10 µg/mL freeze-dried Açaì extract showed a rate constant of SO_4_^2−^ uptake not significantly different with respect to the control (Appendix A). On the contrary, the rate constant value in erythrocytes treated with 50 or 100 mM d-Gal (0.111/0.113 ± 0.001 min^−1^) was significantly increased with respect to the control (*** *p* < 0.001).

In erythrocytes pre-incubated with 0.5 and 10 µg/mL of freeze-dried Açaì extract and then exposed to 50 mM d-Gal, the rate constant (0.058 and 0.055 ± 0.001 min^−1^) was significantly lower than that of erythrocytes treated with 50 mM d-Gal (0.111 ± 0.001 min^−1^), but was not significantly different with respect to the control (Table 2). In erythrocytes pre-incubated with 0.5 or 10 µg/mL of freeze-dried Açaì extract and then exposed to 100 mM d-Gal, the rate constant (0.058 and 0.080 ± 0.001 min^−1^) was significantly lower than that of erythrocytes treated with 100 mM d-Gal (0.113 ± 0.001 min^−1^), but was not significantly different with respect to the control (Table 2). SO_4_^2−^ uptake was almost completely blocked by 10 µM DIDS applied at the beginning of incubation in the SO_4_^2−^ medium (0.017 ± 0.001 min^−1^, *** *p* < 0.001, Table 2). Additionally, the SO_4_^2−^ amount internalized by 50 or 100 mM d-Gal-treated erythrocytes after 45 min of incubation in the SO_4_^2−^ medium was not significantly different compared to the control (Table 2). In DIDS-treated cells, the SO_4_^2−^ amount internalized (5.39 ± 2.50) was significantly lower than that determined in both the control or treated erythrocytes (*** *p* < 0.001, Table 2).

## 4. Discussion

In recent years, an increasing body of research has focused on natural antioxidants and their ability to counteract OS-induced pathological conditions. In fact, the supplementation of phytochemicals in one’s diet has been found to provide numerous health benefits [25,43,44]. The Açai berry represents a good supplementation candidate since it shows several beneficial activities, including antioxidant properties in various experimental models [45,46,47]. In this regard, the present study describes the important protective effects of freeze-dried Açaì extract in a d-Gal-induced aging model in human erythrocytes. This cell-based model has been validated in a previous study, which demonstrates that d-Gal is an efficient tool to induce chronic OS and accelerated aging in erythrocytes [26]. The results shown here indicate that the chemical matrix of Açaì fruit could improve both the structure and function of human erythrocytes as well as pathological events attributable to aging.

There are many studies describing the multiple activities of flavonoids, however, the effects of Açaì fruits on human erythrocytes have not yet been evaluated. Thus, the first step of this research was to test different concentrations of Açaí extract (from 0.005 µg/mL to 1000 µg/mL) to exclude a possible haemolytic power, as well as the ability to induce lipoperoxidation and/or oxidation of proteins. As shown in the Figure 1, 1000 µg/mL freeze-dried Açaì extract significantly increased the percentage of haemolysis after 1 h of incubation compared to cells left untreated. Vice versa, concentrations ranging from 0.005 µg/mL to 500 µg/mL did not cause haemolytic events (Figure 1). Similarly, in erythrocytes treated with increasing concentrations of Açaì extract (from 50 µg/mL to 1000 µg/mL), the estimation of TBARS levels as well as the total sulfhydryl group content were statistically increased compared to those of erythrocytes left untreated. These findings denote that antioxidant compounds in excessive amounts can manifest pro-oxidant activity, in agreement with other studies [48]. On the contrary, lower concentrations (from 0.005 µg/mL to 10 µg/mL) of Açaì extract induced neither an increase in TBARS levels nor in the oxidation of the total sulfhydryl groups, respectively (Figure 2A,B). Based on these findings, 0.5 and 10 µg/mL freeze-dried Açaí extract were selected to be tested for their possible beneficial activity in our OS-related model of aging. These concentrations were in accordance with former studies performed on different cellular types [49,50].

The susceptibility of erythrocytes to d-Gal exposure was investigated in terms of morphological alterations by scanning electron microscopy (SEM). The images (Figure 3) revealed dramatic changes in the erythrocyte shape. The typical biconcave shape was lost in a significant number of cells, which showed surface blebs (acanthocytes, 50 mM d-Gal), or a flattened shape (leptocytes, 100 mM d-Gal). Based on these results, it can be assumed that erythrocytes may undergo dose-dependent damage through characteristic changes in shape. In fact, the percentage of acanthocytes was higher (15.7%) with respect to leptocytes (8.9%) in 50 mM d-Gal-treated erythrocytes (Table 1), while the percentage of leptocytes was higher (28.2%) with respect to acanthocytes (11.4%) in 100 mM d-Gal-treated erythrocytes (Table 1). However, pre-treatment with freeze-dried Açaì extract attenuated the morphological changes (Figure 3), with a reduction of the percentage of both acanthocytes and leptocytes. The study of erythrocyte morphology is of great importance in the field of hemorheology. The morphology of the circulating cells has a fundamental influence on the rheological properties of the blood, and changes in morphology can lead to decreased deformability and increased aggregation [51,52]. These cells may respond to any form of insult by changing their morphology following changes in their membrane or biochemical composition. Certain phenomena such as the oxidation of sulfhydryl groups of membrane proteins, oxidation of membrane fatty acid residues, or oxidation of haemoglobin could alter membrane properties and the cell shape. Since the oxidation of biological macromolecules, such as lipids and proteins, emanates from the deleterious effects of ROS generated during cellular metabolism, intracellular ROS levels have firstly been evaluated. Our findings show that pre-treatment with freeze-dried Açaí (0.5 or 10 µg/mL) extract induced a reduction of ROS production caused by 50 or 100 mM d-Gal incubation (Figure 4A,B). This evidence is supported by many authors and suggests that dietary supplementation with polyphenols and phytochemicals has antioxidant activity, which is generally attributed to their ability to directly neutralise ROS [53,54].

To better explore the oxidation of biological macromolecules, the estimation of TBARS levels—a marker of lipid peroxidation—as well as the sulfhydryl group content of the total proteins have been considered, respectively. Our results show that 1 h pre-treatment with 0.5 and 10 µg/mL freeze-dried Açaì extract avoided the lipid peroxidation of membranes induced by treatment with 50 or 100 d-Gal (Figure 5A,B). Similarly, the measurement of the sulfhydryl group content was also evaluated. Açaì extract (0.5 and 10 µg/mL) also protected erythrocyte proteins from oxidative damage (Figure 6A,B). In a previous investigation [55], we evaluated the potential protective role of quercetin (Q), a polyphenolic flavonoid compound, on an aging model represented by human erythrocytes treated with 100 mM d-Gal for 24 h. In this regard, TBARS levels, which were augmented following 100 mM d-Gal exposure (24 h), were completely restored by 10 µM Q pre-treatment (1 h). Similarly, the freeze-dried Açaì extract pre-treatment (0.5–10 µg/mL for 1 h) completely restored TBARS levels in this experimental model. On the contrary, the oxidation of protein sulfhydryl groups was partially restored after pre-treatment (1 h) with Q or Açaì extract in erythrocytes treated with 100 mM d-Gal (24 h). Although the activity of these two antioxidants can depend on their concentration as well as phytochemical components, we can conclude that Q and Açaì extract show a similar antioxidant power in our experimental model and could, therefore, play a crucial role in counteracting oxidative stress increases in erythrocytes. These data are in line with what has been previously demonstrated by other authors. For example, the effects of Açaì have been studied on anaerobic exercise-induced changes in the blood antioxidant defense system in junior hurdlers. Specifically, six weeks of daily consumption of a diet supplemented with Açaì pulp resulted in significant increases in plasma total polyphenols and blood GSH content compared to baseline levels, as well as beneficial changes in the lipid profile (TBARS levels) [56]. In addition, experiments in rats revealed that a diet supplemented with 2% Açai pulp for six weeks caused a reduction in erythrocyte protein oxidation compared to control animals [57]. In fact, compared with other antioxidant-rich fruits, Açaì extract has 4.8, 6.1, and 7.5-fold the antioxidant capacity of blackberries, blueberries, and strawberries, respectively [23]. The antioxidant effect of Açaì extract has been attributed to its phytochemical composition comprising hydroxybenzoic acids and flavanols, along with cyanidin 3-O-rutinoside and cyanidin 3-O-glucoside as the predominant anthocyanins [58].

Traditionally, erythrocyte phagocytosis has been proposed to be the result of the accumulation of “eat me” signals on the membrane of aged erythrocytes. Several “eat me” signals have been identified to be important for the clearance of aged erythrocytes by macrophages residing in the spleen, or alternatively, in the liver [59]. Aging has been associated with eryptosis, a controversial process that closely mimics the programmed cell death of nucleated cells (apoptosis). This phenomenon is characterized by a progressive increase in membrane phospholipid asymmetry, owing to the consumption of ATP reservoirs, which results in the apoptosis-like externalization of PS to the outer leaflet of the plasma membrane [60,61]. The externalization of PS on the surface of eryptotic cells can have two major pathophysiological implications: on one hand, it initiates the phagocytosis of erythrocytes; on the other hand, it mediates the adherence of erythrocytes to vascular endothelium cells, which also express PS receptors. Excessive eryptosis initiating the phagocytosis of many red blood cells may, therefore, result in the acute loss of erythrocytes. Exposure to 50 and 100 mM d-Gal did not induce the translocation of PS at the outer plasma membrane leaflet (Figure 7), thus demonstrating that, in this aging model, erythrocytes remain in an early phase of the aging process. A similar condition could be detected in OS-related pathologies, including anaemia, cardiovascular complications, chronic kidney disease, and diabetes [62,63]. Hence, the beneficial effects of Açaì extract shown here apply to an early stage of aging.

In human erythrocytes, CD47 protein is part of the mechanisms regulating membrane architecture, but can also function as an important marker of “self” [64]. In fact, it has been demonstrated that the expression of CD47 on the red blood cell surface could be one of the mechanisms that regulate the removal of senescent cells from the blood flow by phagocytosis. Treatment with 50 mM d-Gal for 24 h did not induce a dramatic loss of this protein, but rather its rearrangement and redistribution (Figure 8E), which could be linked to the progressive plasma membrane blebbing and vesiculation shown by SEM images (Figure 3). Instead, 100 mM d-Gal treatment for 24 h induced both a loss and a redistribution of CD47 expression on the plasma membrane (Figure 8E). This latter finding was paralleled by the appearance of cells with an atypical shape, identified as leptocytes, in SEM images (Figure 3).

During their lifetime, erythrocytes release both exosomes and plasma membrane-derived ectosomes. During the development of erythrocytes, exosomes are generated through a process that involves the double invagination of the endosomal membrane to form multivesicular bodies containing intraluminal vesicles. This process is followed by the fusion of the multivesicular bodies to the plasma membrane to produce exosomes. By contrast, ectosomes (micro-vesicles) and large vesicles are generated by the outward budding of the plasma membrane, followed by vesicle shedding [65,66]. Vesiculation is a way to get rid of dangerous molecules, such as oxidized proteins or denatured haemoglobin. Multilevel changes in lipids and proteins of erythrocyte membranes, caused by aging, cause a decrease in erythrocyte deformability and unfavourable changes in blood flow, which promote additional OS, proneness to atherosclerotic lesions, and an increase in blood viscosity. As a result of ectosome formation, the protein composition of erythrocytes varies among circulating erythrocytes, with new erythrocytes being larger with a full complement of membrane proteins and old erythrocytes smaller and denser with a significantly lower content in membrane proteins, including CD47 and B3p [67].

In contrast to the “eat me” signals, CD47 could inhibit phagocytosis of erythrocytes by macrophages during early aging. It is probable that CD47 exerts its inhibitory effect through binding to the signal regulatory protein (SIRP) alpha on macrophages, which induces inhibitory signaling by the immunoreceptor tyrosine-based inhibition motifs (ITIMs) residing in the cytoplasmic tail of the SIRP alpha. Upon the binding of CD47 to the SIRP alpha, the tyrosine phosphatases SHP-1 and SHP-2 are recruited to the ITIMs and activated, which in turn regulates downstream signaling pathways and effector functions, generally in a negative fashion [59]. However, a total loss of CD47 expression associated to PS externalisation, as well as the clustering of the extracellular regions of B3p, might be proposed as a removal mechanism of aging erythrocytes. In addition, the potential conversion of CD47 from a “do not eat me” to an “eat me” signal is due to a conformational change in CD47. Intriguingly, erythrocyte phagocytosis after this switch seems to be mediated by the same receptor that normally signals the inhibition of phagocytosis, that is, the SIRP alpha [68]. In our model of aging, CD47 expression and distribution were completely re-established in erythrocytes pre-incubated with 0.5 and 10 µg/mL of freeze-dried Açaí extract. This evidence is also confirmed by a recovery of erythrocyte morphological properties in SEM images (Figure 3, Table 1).

As mentioned above, CD47 can be modified by an OS increase, or alternatively, changes in the organisation of the B3p complex; both are potential extracellular indicators of intracellular damage and aging in human erythrocytes [68]. To better clarify the role of B3p, the expression of this protein was also investigated. Specifically, data show that B3p re-arranges in surface blebs, favouring its reorganisation under 50 mM d-Gal treatment (Figure 9E). Vice versa, the treatment with 100 mM d-Gal induced both a loss and a redistribution of the protein, most probably caused by vesicle shedding [67] (Figure 9E).

Band 3 protein is an integral membrane protein that accounts for approximately 25% of the erythrocyte membrane surface. It has several crucial functions and it is composed by two rather different domains of a similar size [69]. The C-terminal domain mediates the chloride–bicarbonate Cl^−^/HCO_3_^−^ exchange across the plasma membrane [70,71]. Instead, the N-terminal cytoplasmic domain contains binding sites for cytoskeletal and cytoplasmic proteins, including haemoglobin [72,73]. Haemoglobin glycation is the first example of the non-enzymatic glycation of proteins, which can contribute to the development of complications in various diseases associated with aging [74]. To better study the molecular interaction between B3p and haemoglobin, A1c levels were evaluated. Our results indicated that exposure to 50 or 100 mM d-Gal for 24 h increased the content of A1c levels (Figure 10A,B). These modifications can initiate a cascade of biochemical and structural transformations, including the clustering of B3p regions. When aging processes are advanced, these clusters could provide a recognition site for antibodies directed against aging cells, thus triggering the premature removal of senescent erythrocytes from the circulation at the end of their 120-days life span. Pre-treatment with 0.5 or 10 µg/mL of Açaí extract prevented A1c formation (Figure 10A,B).

At last, our focus has been addressed to assay the anion exchange capability through B3p [15,75,76,77]. Hence, the SO_4_^2−^ uptake was measured after treatment with d-Gal (50 or 100 mM) with or without pre-exposure to 0.5 or 10 µg/mL of freeze-dried Açaí extract [26,29]. In erythrocytes incubated with d-Gal (50 and 100 mM), the rate constant for SO_4_^2−^ uptake was accelerated compared to the control (Figure 11A,B, Table 2). However, one-hour pre-treatment with 0.5 or 10 µg/mL of Açaí extract completely restored the rate constant of SO_4_^2−^ uptake, which demonstrated a protective effect of the Açaí extract on B3p function. Previously, we reported that the exposure of erythrocytes to 10 mM d-Gal for 24 h induced a reduction of the rate constant of SO_4_^2−^ uptake, which was not, however, associated with an OS increase [78]. Therefore, the different effect on the SO_4_^2−^ transport kinetics observed in the present study is not surprising. Indeed, a reduction of the transport rate in a former study could be most likely linked to mechanisms other than OS, putatively, to the formation of glycated haemoglobin. The evidence that B3p exhibits modifications in the rate constant for SO_4_^2−^ uptake following the exposure of human erythrocytes to OS has also been previously demonstrated. Specifically, H_2_O_2_-induced OS provoked a reduction in the rate constant for SO_4_^2−^ uptake [33,79], whereas treatment with high glucose concentrations induced an acceleration of the anion exchange [29]. Therefore, it is tempting to speculate that such a two-sided effect on anion exchange velocity depends on the specific structure targeted by the stressors and the underlying pathways.

## 5. Conclusions

We conclude that exposure to 50 or 100 mM d-Gal induced aging in human erythrocytes, and manifested as increased OS, dose-dependent distinct morphological changes, as well as an alteration of B3p and CD47 protein surface expression. Pre-treatment with 0.5 or 10 µg/mL Açaí extract avoided the formation of acanthocytes or leptocytes observed after exposure to 50 and 100 mM d-Gal, respectively, prevented d-Gal-induced OS damage, including ROS production, lipid peroxidation, as well as sulfhydryl group oxidation of total proteins, and restored the distribution of B3p and CD47 on the plasma membrane. Moreover, d-Gal exposure was associated with an acceleration of the rate constant of SO_4_^2−^ uptake through B3p, as well as A1c formation. Both alterations have been attenuated by pre-treatment with the Açaì extract. The present study provides mechanistic insights into the benefits of polyphenol-rich extracts on cells exposed to OS. In this light, future investigations are needed to clarify the signaling underlying restoration of a normal anion exchange ability by the Açaí extract, including possible effects on the interaction between B3p and the cytoskeletal proteins ankirin and spectrin, their potential post-translation modifications, as well as a possible influence on the endogenous antioxidant machinery.

## Figures and Tables

**Figure 1 cells-11-02391-f001:**
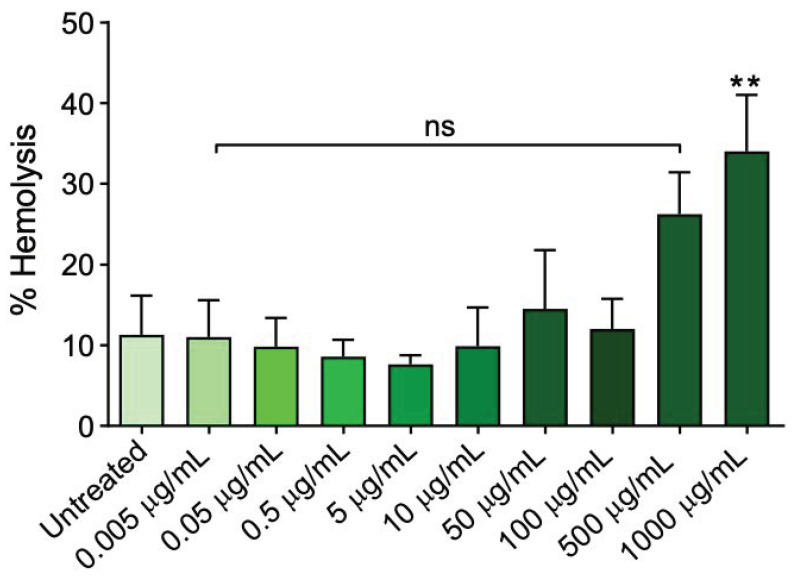
Haemolysis measurement. Human erythrocytes have been exposed to increasing concentrations of freeze-dried Açaì extract for 1 h at 37 °C. ns, not statistically significant versus control; **, *p* < 0.01 versus control, ANOVA with Dunnet’s post-test (*n* = 10).

**Figure 2 cells-11-02391-f002:**
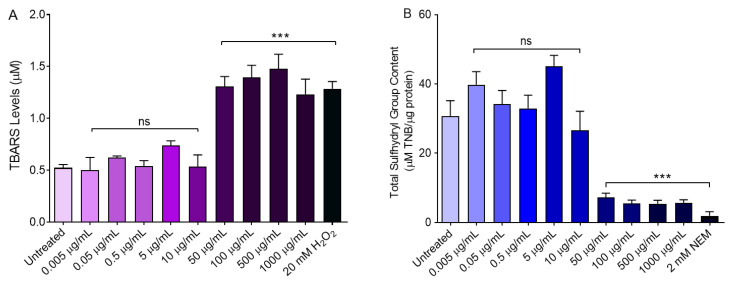
Oxidative stress assessment. Human erythrocytes have been exposed to increasing concentrations of freeze-dried Açaì extract for 1 h at 37 °C. (**A**) Estimation of TBARS levels—a marker of lipid peroxidation—as well as (**B**) total sulfhydryl group content (µM TNB/µg protein). ns, not statistically significant versus control; ***, *p* < 0.001 versus control, ANOVA with Dunnet’s post-test (*n* = 10).

**Figure 3 cells-11-02391-f003:**
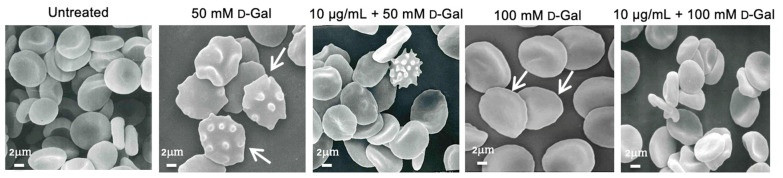
Erythrocyte morphology evaluation. Representative scanning electron microscopy images showing erythrocytes with a typical biconcave form (left untreated); with surface blebs (acanthocytes, arrows) (50 mM d-Gal); with a flattened shape (leptocytes, arrows) (100 mM d-Gal). Pre-treatment with freeze-dried Açaì extract (10 µg/mL) attenuated the morphological changes compared to d-Gal treatment. Magnification 3000×.

**Figure 4 cells-11-02391-f004:**
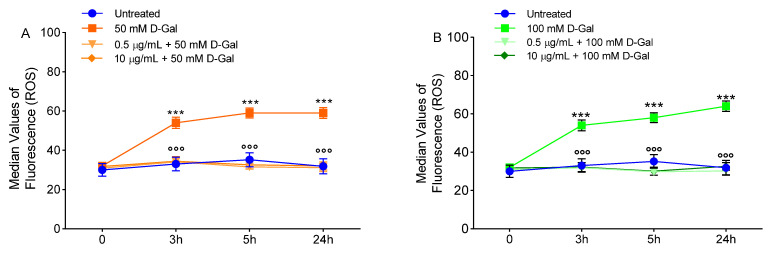
Detection of reactive oxygen species (ROS) levels by flow cytometry. Time course of ROS production in erythrocytes left untreated (control) or treated for 24 h with 50 (**A**) or 100 mM (**B**) d-Gal, with or without pre-exposure to 0.5 or 10 µg/mL freeze-dried Açaì extract for 1 h. ns, not statistically significant versus control; ***, *p* < 0.001 versus control; ^°°°^, *p* < 0.001 versus d-Gal, one-way ANOVA followed by Bonferroni’s post hoc test (*n* = 8).

**Figure 5 cells-11-02391-f005:**
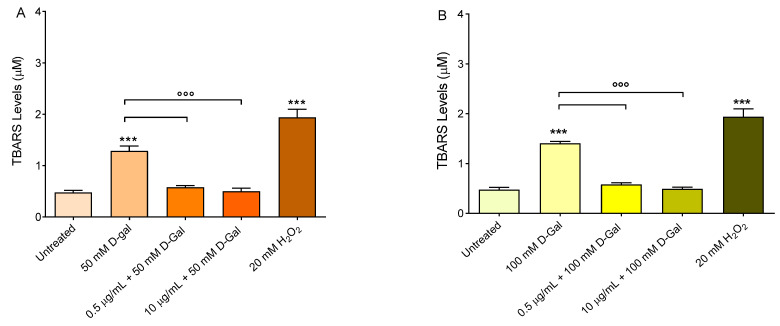
Detection of TBARS levels. TBARS levels (µM) in erythrocytes left untreated (control) or treated for 24 h with 50 (**A**) or 100 mM (**B**) d-Gal, with or without pre-incubation for 1 h with 0.5 or 10 µg/mL freeze-dried Açaì. H_2_O_2_ (20 mM, 1 h at 37 °C) was used as the positive control. ***, *p* < 0.001 versus control; ^°°°^, *p* < 0.001 versus 50 and 100 mM d-Gal, one-way ANOVA followed by Bonferroni’s post hoc test (*n* = 10).

**Figure 6 cells-11-02391-f006:**
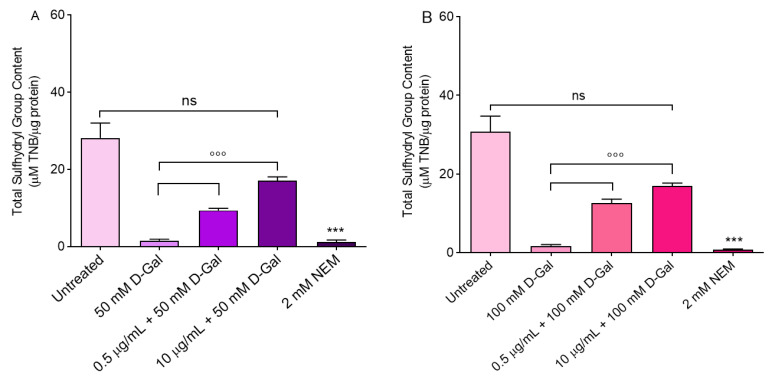
Measurement of total SH group content. Sulfhydryl group content (µM TNB/µg protein) in erythrocytes left untreated (control) or treated with 50 (**A**) or 100 (**B**) mM d-Gal for 24 h with or without pre-incubation for 1 h with 0.5 or 10 µg/mL freeze-dried Açaì extract, or alternatively, with 2 mM NEM (positive control). ns, not statistically significant versus control; °°°, *p* < 0.001 versus 50 and 100 mM d-Gal; ***, *p* < 0.001 versus control, one-way ANOVA followed by Bonferroni’s multiple comparison post-hoc test (*n* = 10).

**Figure 7 cells-11-02391-f007:**
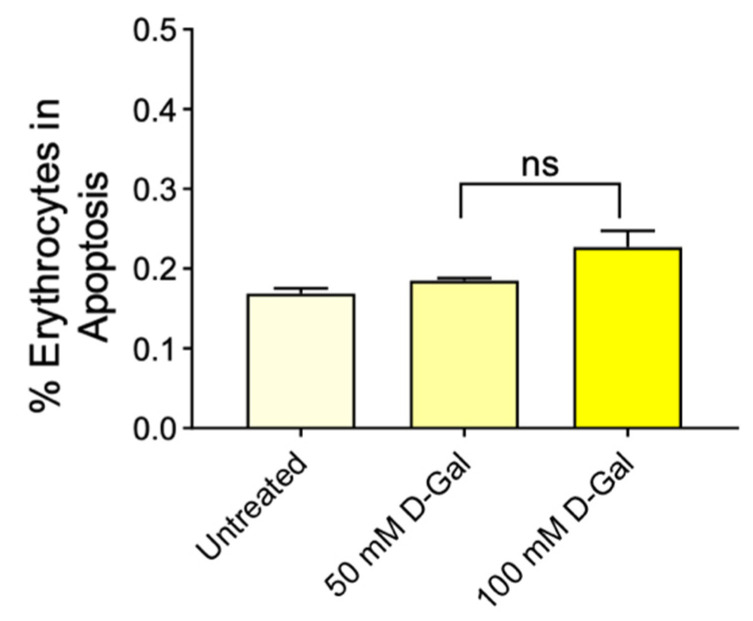
Detection of apoptosis by flow cytometry. Percentage of apoptotic erythrocytes positive to Annexin 5 and Trypan blue detected in samples treated with 50 and 100 mM d-Gal. ns, not statistically significant versus control, ANOVA with Dunnet’s post-test (*n* = 8).

**Figure 8 cells-11-02391-f008:**
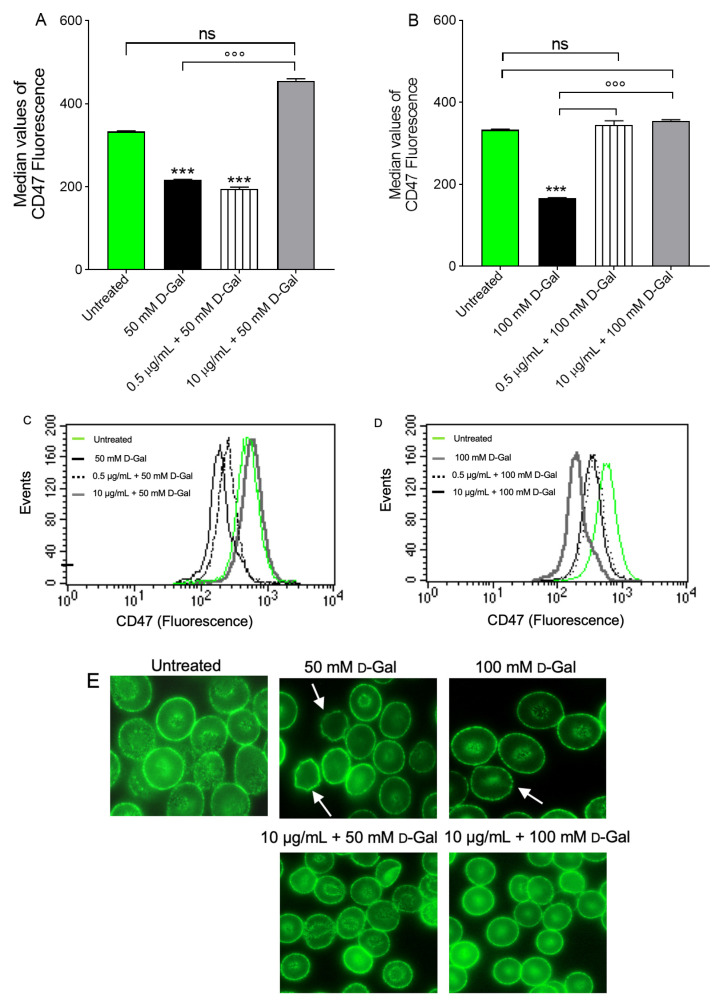
Flow cytometry immunofluorescence of CD47 protein expression. Erythrocytes were treated for 24 h with 50 (**A**) or 100 mM (**B**) d-Gal, with or without pre-incubation for 1 h with 0.5 or 10 µg/mL freeze-dried Açaì extract. Histograms represent median values of fluorescence intensity. In (**C**,**D**), typical flow cytometry measurements of CD47 expression of a representative experiment are shown. In (**E**), representative images of CD47 expression obtained by flow cytometry immunofluorescence are shown. Samples were observed with a 100× objective. Note the significant morphological changes in both 50 and 100 mM d-Gal (arrows). ns, not statistically significant versus control; ***, *p* < 0.001 versus control; °°°, *p* < 0.001 versus 50 or 100 mM d-Gal, one-way ANOVA followed by Bonferroni’s post hoc test (*n* = 8).

**Figure 9 cells-11-02391-f009:**
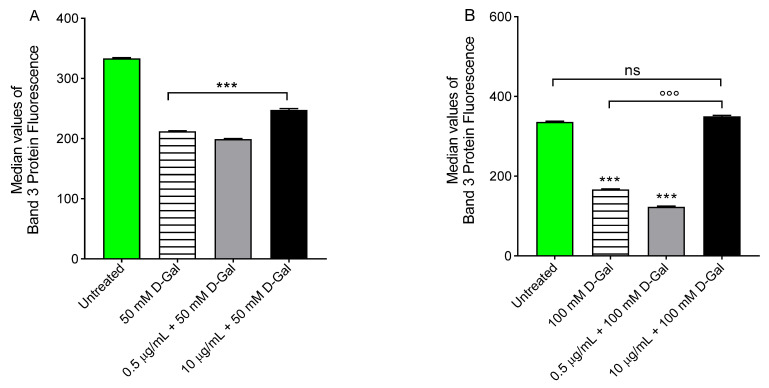
Flow cytometry immunofluorescence of B3p protein expression. Erythrocytes were treated for 24 h with 50 (**A**) or 100 mM (**B**) d-Gal, with or without pre-incubation for 1 h with 0.5 or 10 µg/mL freeze-dried Açaì extract. Histograms represent median values of fluorescence intensity. In (**C**,**D**), typical flow cytometry measurements of B3p expression of a representative experiment are shown. In (**E**), representative micrographs obtained by flow cytometry immunofluorescence showing B3p distribution in erythrocytes left untreated, treated with 50 mM or 100 mM d-Gal, or alternatively, pre-treated with 10 µg/mL freeze-dried Açaì extract, and then exposed to 100 mM d-Gal are shown. Samples were observed with a 100x objective. Note the significant morphological changes in both 50 and 100 mM d-Gal (arrows). ns, not statistically significant versus untreated; ***, *p* < 0.001 versus untreated (control); °°°, *p* < 0.001 versus 100 mM d-Gal, one-way ANOVA with Bonferroni’s multiple comparison post-hoc test (*n* = 8).

**Figure 10 cells-11-02391-f010:**
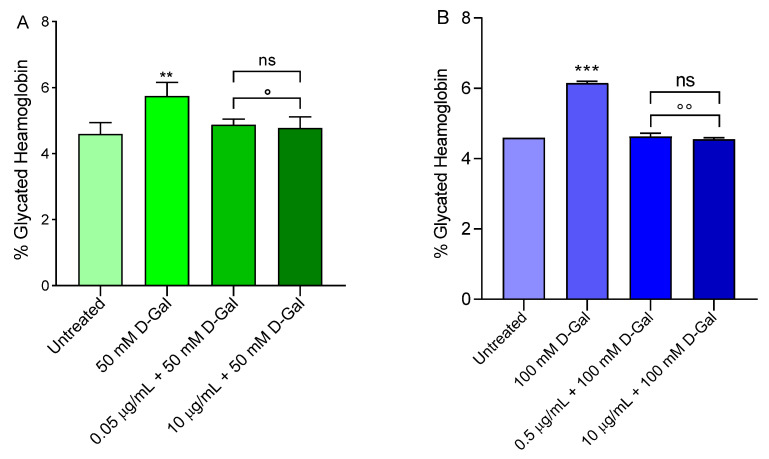
Glycated haemoglobin content (%A1c). Erythrocytes were left untreated or incubated for 24 h with 50 (**A**) or 100 mM (**B**) d-Gal, with or without pre-exposure to 0.5 or 10 µg/mL freeze-dried Açaì extract (pre-incubation for 1 h). ns, not statistically significant versus untreated; **, ***, *p* < 0.01, and *p* < 0.001 versus untreated (control); °, *p* < 0.05 versus 50 mM d-Gal; °°, *p* < 0.01 versus 100 mM d-Gal, one-way ANOVA with Bonferroni’s multiple comparison post-hoc test (*n* = 10).

**Figure 11 cells-11-02391-f011:**
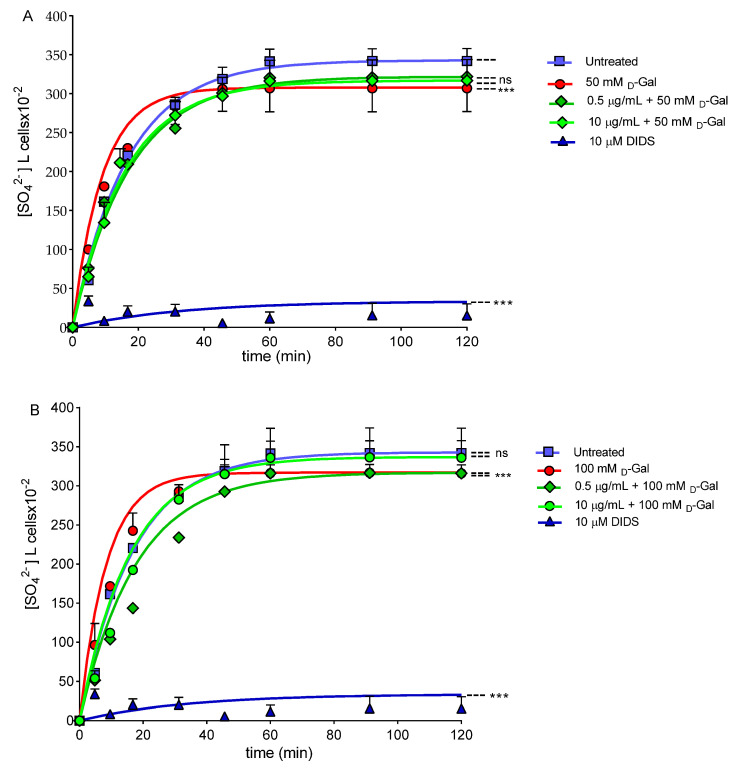
Time course of SO_4_^2−^ uptake. Erythrocytes were left untreated (control) or treated with (**A**) 50 or (**B**) 100 mM D-Gal with or without pre-exposure to 0.5 or 10 µg/mL freeze-dried Açaì extract (pre-incubation for 1 h), or 10 µM DIDS. ns, not statistically significant versus control; ***, *p* < 0.001 versus control, one-way ANOVA followed by Bonferroni’s post hoc test.

**Table 1 cells-11-02391-t001:** Percentage of morphological alterations in erythrocytes left untreated (control) or treated as indicated. Data are presented as means ± S.E.M. from separate three independent experiments, where ^ns^, not statistically significant versus untreated (biconcave shape, acanthocytes, and leptocytes); **, *p* < 0.01 versus control (biconcave shape); ^°°^ *p* < 0.01 versus 50 or 100 mM d-Gal (biconcave shape). ^^^^^, *p* < 0.001 versus control (acanthocytes); ^$$^, ^$$$^, *p* < 0.01 and *p* < 0.001 versus 50 or 100 mM d-Gal (acanthocytes). ^£^, ^£££^, *p* < 0.05, and *p* < 0.001 versus control (leptocytes); ^çç^, ^ççç^, *p* < 0.01 versus 100 mM d-Gal (leptocytes); one-way ANOVA followed by Bonferroni’s multiple comparison post-hoc test.

	Biconcave Shape	Acanthocytes	Leptocytes
**Untreated (control)**	84% ± 0.014	6% ± 0.012	5% ± 0.010
**50 mM d-Gal**	80% ± 0.011 ^ns^	15.7% ± 0.011 ^^^^^	8.9% ± 0.011 ^£^
**0.5 µg** **/mL + 50 mM d-Gal**	88.8% ± 0.010 ^ns^	6.9% ± 0.008 ^ns,$$$^	4.3% ± 0.009 ^ns^
**10 µg** **/mL + 50 mM d-Gal**	86.3% ± 0.009 ^ns^	9.8% ± 0.009 ^ns,$$$^	3.9% ± 0.009 ^ns^
**100 mM d-Gal**	60.4% ± 0.008 **	11.4% ± 0.007 ^^^^^	28.2% ± 0.010 ^£££^
**0.5 µg** **/mL + 100 mM d-Gal**	69.3% ± 0.012 **	8.7% ± 0.0012 ^ns,$$^	22% ± 0.011 ^çç^
**10 µg** **/mL + 100 mM d-Gal**	80% ± 0.0011 ^ns,°°^	8% ± 0.0012 ^ns,$$^	12% ± 0.014 ^ççç^

**Table 2 cells-11-02391-t002:** Rate constant of SO_4_^2−^ uptake and amount of SO_4_^2−^ trapped in erythrocytes left untreated (control) and erythrocytes treated as indicated. Data are presented as means ± S.E.M. from separate (*n*) experiments, where ^ns^, not statistically significant versus untreated; ***, *p* < 0.001 versus control; ^°°°^, *p* < 0.01 versus 50 or 100 mM d-Gal, one-way ANOVA followed by Bonferroni’s multiple comparison post-hoc test.

Experimental Conditions	Rate Constant (min^−1^)	Time (min)	n	SO_4_^2−^ Amount Trapped after 45 min of Incubation in SO_4_^2−^ Medium [SO_4_^2−^] l Cells × 10^−2^
**Untreated (control)**	0.059 ± 0.001	16.67	10	318.60 ± 18.63
**50 mM d-Gal**	0.111 ± 0.002 ***	8.95	10	306.35 ± 19.90 ^ns^
**0.5 µg/mL Açaì Extract + 50 mM d-Gal**	0.058 ± 0.001 ^ns,°°°^	17.09	10	301 ± 15.90 ^ns^
**10 µg/mL Açaì Extract + 50 mM d-Gal**	0.055 ± 0.001 ^ns,°°°^	17.83	10	314 ± 15.80 ^ns^
**100 mM d-Gal**	0.113 ± 0.001 ***	8.75	9	306.35 ± 16.50 ^ns^
**0.5 µg/mL Açaì Extract + 100 mM d-Gal**	0.056 ± 0.001 ^ns,°°°^	17.70	9	301 ± 19.80 ^ns^
**10 µg/mL Açaì Extract + 100 mM d-Gal**	0.062 ± 0.001 ^ns,°°°^	15.79	8	349.97 ± 11.15 ^ns^
**10 µM DIDS**	0.017 ± 0.001 ***	62.50	16	5.49 ± 3.50 ***

## Data Availability

The data that support the findings of this study are available from the corresponding author upon reasonable request.

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
