# Peer review of "Açaì (Euterpe oleracea) Extract Protects Human Erythrocytes from Age-Related Oxidative Stress"

_cells, 2022, doi:10.3390/cells11152391_

Round 1

Reviewer 1 Report

The manuscript by Remigante et al., show the protective role of Açaì extract in a D-Galactose (D-Gal)-induced model of aging in human erythrocytes. Authors use several in vitro approaches to demonstrate the effect of Acai extract on ROS production, thiobarbituric acid reactive substances (TBARS) levels, oxidation of protein sulfhydryl groups, anion exchange capability through Band 3 protein (B3p) and glycated haemoglobin (A1c). Abstract is clear and concise; titles are informative and reflects the description. Experiments are well designed, executed and the conclusions drawn are supported by the results. Overall, I recommend this manuscript for publication.

Author Response

Point by point reply to the Reviewer 1
The manuscript by Remigante et al., show the protective role of Açaì extract in a D-Galactose (D-Gal)-induced model of aging in human erythrocytes. Authors use several in vitro approaches to demonstrate the effect of Acai extract on ROS production, thiobarbituric acid reactive substances (TBARS) levels, oxidation of protein sulfhydryl groups, anion exchange capability through Band 3 protein (B3p) and glycated haemoglobin (A1c). 
Abstract is clear and concise; titles are informative and reflects the description. Experiments are well designed, executed and the conclusions drawn are supported by the results. Overall, I recommend this manuscript for publication. We thank the Reviewer for the overall positive evaluation

Reviewer 2 Report

This study aimed to verify the protective role of Açaì extract in a D-Galactose (D-Gal)-induced model of aging in human erythrocytes. Lots of work has been done. However, there are some minor mistakes and suggestions to the authors.

1.     The information of the extract was missed. Related information of the content for main ingredient should be provided such as flavonoids, polyphenols, anthocyanins et al. In theory, the mass spectrometry was needed.

2.     It is recommended to add positive drugs during the experiment, so as to better characterize the efficacy of the extract.

3.     The flow cytometry of figure 7 must be provided.

4.     The results from figures, conducted with fluorescent probes must be quantified in graphs and not just representative microphotographs.

5.     The scale bar of figures is missing in the photos.

Author Response

Point by point reply to the Reviewer 2
This study aimed to verify the protective role of Açaì extract in a D-Galactose (D-Gal)-induced model of aging in human erythrocytes. Lots of work has been done. However, there are some minor mistakes and suggestions 
to the authors. We thank the Reviewer for the overall positive evaluation.
1. The information of the extract was missed. Related information of the content for main ingredient should be provided such as flavonoids, polyphenols, anthocyanins et al. In theory, the mass spectrometry was needed.
We want to thank the Reviewer for raising this point. A sentence has been added in the main text in this regard.
The freeze-dried Açaì extract (CAS Number: 879496-95, 906351-38-0) was originally purchased from 
Farmalabor Srl (Canosa di Puglia, Barletta, Italy) and then kindly provided to us by Professor M. Cordaro 
from the University of Messina. Powder extract with 10% polyphenols was previously analyzed by a spectrophotometric method. We have attached the technical datasheet.
2. It is recommended to add positive drugs during the experiment, so as to better characterize the efficacy of the extract. We want to thank the Reviewer for raising this point. During experimental design, using positive 
drugs was not planned. The freeze-dried Açaì extract is composed of a mixture of bioactive substances, including numerous phytochemicals components, such as flavonoids. Thus, the effects of the extract cannot be compared to a single molecule. However, in a previous investigation (DOI: 10.3390/ijms23147781), we 
evaluated the potential protective role of Quercetin, a polyphenolic flavonoid compound (CAS Number: 117-39-5) on an aging model represented by human erythrocytes treated with 100 mM D-Galactose for 24 hours.
In this regard, TBARS levels, a marker of lipid-peroxidation, which were augmented following 100 mM DGalactose exposure (24 hours), were completely restored by 10 µM Quercetin pre-treatment (1 hour). Similarly, the freeze-dried Açaì extract pre-treatment (0.5-10 µg/ml for 1 hour) completely restored TBARS levels in the same experimental model. On the contrary, oxidation of protein sulfhydryl groups was partially restored after pre-treatment (1 hour) with Quercetin or Açaì extract in erythrocytes treated with 100 mM D-Galactose (24 
hours). Although the activity of these two antioxidants can depend on their concentration as well as phytochemical components, we can conclude that quercetin and Açaì extract show similar antioxidant power in our experimental model and could therefore play a crucial role in counteracting oxidative stress increase in erythrocytes.
3. The flow cytometry of figure 7 must be provided. We thank the reviewer for this suggestion. This graph 
actually represents the detection of apoptosis by flow cytometry. We have modified the figure legend for better 
clarity.
4. The results from figures, conducted with fluorescent probes must be quantified in graphs and not just 
representative microphotographs. We want to thank the Reviewer for raising this point. However, in figures 8 and 9, histograms of panels A and B related to B3p and CD47 expression indicate the median values of 
fluorescence intensity and correspond to the microphotographs obtained by flow cytometry immunofluorescence shown in panels F. In Figures 8 and 9, the figure legend was also modified for better clarity.
5. The scale bar of figures is missing in the photos. We thank the reviewer for this suggestion. A scale bar forFigure 3 was added. In Figures 8 and 9, the magnification (100x objective) is indicated in the figure legend.

Reviewer 3 Report

The authors have checked if extracts of Acai protect human erythrocytes from oxidative stress, using a variety of methods. The work is well-rounded and scientifically sound. It is recommended for publication.

Author Response

Point by point reply to the Reviewer 3
The authors have checked if extracts of Acai protect human erythrocytes from oxidative stress, using a variety of methods. The work is well-rounded and scientifically sound. It is recommended for publication. We thank 
the Reviewer for the overall positive evaluation.